# Victory Tax: A Holistic Income Tax System

**DOI:** 10.3390/e23111492

**Published:** 2021-11-11

**Authors:** Donald J. Jacobs

**Affiliations:** Department of Physics and Optical Science, University of North Carolina at Charlotte, Charlotte, NC 28223, USA; djacobs1@uncc.edu

**Keywords:** income tax, tax deduction, income redistribution, government transfer, government dependency, poverty line, basic income guarantee, effective tax rate, balanced budget, elastic tax

## Abstract

How can an income tax system be designed to exploit human nature and a free market to create a poverty free society, while balancing budgets without disproportional tax burdens? Such a tax system, with universal character, is deduced from the following guiding principles: (1) a single tax rate applies to all income types and levels; (2) the tax rate adjusts to satisfy budget projections; (3) government transfer only supplements the income of households with self-generated income below the poverty line; (4) deductions for basic living expenses, itemized investments and capital losses are allowed; (5) deductions cannot be applied to government transfer. A general framework emerges with three parameters that determine a minimum allowed tax deduction, a maximum allowed itemized deduction, and a maximum deduction defined by income percentage. An income distribution that mimics the United States, and a series of log-normal distributions are considered to quantitatively compare detailed characteristics of this tax system to progressive and flat tax systems. To minimize government dependency while maximizing after-tax income, the effective tax rate (*ETR*) as a function of income percentile takes the shape of the letter, V, inspiring the name victory tax, where the middle class has the lowest *ETR*.

## 1. Introduction

Many forms of taxation evolved organically in different political and economic systems over human history [1]. A tax system deals with the flow of money from society to government, as tax revenues are collected and given back to society as government spends tax revenue, including government transfers in relation to wealth redistribution. Historically, the extent and purpose of taxation generally was not to benefit society as a whole [1]. Today, opinions on the extent and purpose of taxation are often politically charged, making it impossible to design a tax system that is acceptable to all positions. Arguments from different philosophical perspectives on relationships between society, economy, and public policy identify complex issues that need to be reconciled to achieve a rational tax system. The impetus of this work comes from many laudable debates in the United States (US) about the federal tax system regarding how to set tax rates, and on related issues that affect the nature of taxation that include government spending, short-term deficits, long-term debt, government dependency, poverty, income inequality, disproportionate tax burdens, and the diminishing wealth of the middle class. These ongoing debates indicate that the structure of a tax system has a significant impact on the economy and well-being of society. The aim of this work is to construct the foundation for a holistic tax system with universal appeal divorced from ideology, by taking a pragmatic approach to solving a wide range of problems faced by modern society.

### 1.1. Motivations for a Holistic Tax System with Desirable Characteristics

Based on the Gini index [2] from 1980 to the present, the middle class of the US is steadily shrinking, suggesting that the federal individual income tax could be changed to strengthen the middle class. Although a strong middle class fuels a consumer economy, the distribution of income (and wealth) in free markets is empirically found to be highly skewed toward a tiny percent of its population [3,4,5]. Remarkably, there is universality in heavy tailed distributions for income and wealth across a population in free markets, regardless of government form and tax law. This work accepts the inevitability of highly skewed income dispersion in free-market economies, and then uses the free market and human nature to its advantage.

It is prudent to create incentives for economic growth, while discouraging government dependency, which can lead to complacency and a less productive society. In this context, regressive and progressive taxation affects the economy and redistribution of wealth in different ways. When government redistributes wealth, the effective tax rate (*ETR*) will vary between households. A (higher, lower) *ETR* makes it (more, less) difficult for a household to generate wealth due to a (higher, lower) tax burden. A pragmatic reason for the government to redistribute income is that this practice creates an environment in which all segments of society can accumulate wealth.

A widely employed definition of a flat tax involves taxing labor income at a single marginal tax rate with an allowed deduction [6]. In this work, a flat tax is defined as having the *ETR* independent of income type and level. In practice, a flat tax is achieved by not allowing tax deductions or government transfers, so that no redistribution of income is made. In the deductive approach taken here, the *ETR* dependence on income level is not a priori assumed. Rather, the *ETR* will be a result of a set of guiding principles concerned with fairness, maintaining long-term stability for society, and creating an incentive for personal economic growth at all income levels. There should be a simple way for the government to collect tax revenues without runaway deficits and not impose excessive tax burdens on the population. If a tax system achieves these goals using a strange looking *ETR*, so be it.

A holistic tax system should create a net benefit to society while promoting individual interests, and therefore can be implemented by any political system provided the government is sincere about respecting human dignity and wants to maintain a stable free-market economy. Minimally, all persons in society should have equal opportunity to generate personal wealth from a market economy. The tax system should create incentives for individuals to generate wealth. The mathematical framework of the tax system should not be tied to specific policies. The tax system should have a structure independent of a population’s income distribution. Moreover, it should be easy for a household to determine its tax burden, pay owed taxes, and receive government assistance when needed. Likewise, it should be easy for the government to manage administratively, deter runaway deficits, and adapt to society’s needs. Applied across the income spectrum, the tax system should capitalize on the free market to create income growth opportunities, encourage self-reliance, and minimize government dependence.

### 1.2. Contributions from a Scientific Approach

To my knowledge, the tax system developed herein is novel, although there are similarities with the concepts of a negative income tax [7] and basic income guarantee [8]. Specifically, the principles for a holistic tax system developed in Section 2.1 based on pragmatic considerations identify negative income tax and basic income guarantee as inadmissible. Positive and negative aspects of these other proposed tax systems have been extensively discussed [9,10,11,12,13,14,15,16]. The tax system developed here escapes the pitfalls of negative income tax [17]. The main criticism of a negative income tax is that it removes an incentive for people to work in low-wage jobs when it is more lucrative to obtain greater income when not working. This problem is generally referred to as a welfare trap [18]. Common criticisms against a basic income guarantee are that marginal tax rates become very high [19,20,21,22], it is too risky [19], and it is cost prohibitive on large scales [20,21]. Following five principles that are conceptually justified below, a simple holistic tax system emerges, and it is shown in Section 3 to be feasible and cost-effective.

The concept of a basic income guarantee was discovered independently many times under different names [16]. In this paper, the similar concept of a need-based income guarantee is the result of the development of a pragmatic tax system based on a deductive approach. The critical difference to previously proposed schema [13] is in the way government transfer is handled. After constructing five guiding principles, which include a single tax rate with three types of tax deductions, the derived tax system provides a surprisingly low population average *ETR*, where the most tax relief goes to the middle class. The *ETR* for the ultra-rich is no higher than for the extreme poor. Moreover, low-income households are subjected to a regressive *ETR* that eliminates the welfare trap and encourages the poor to gain financial independence and move into the middle class. The tax system is unified with the welfare system, so that poverty and deficits can be virtually eliminated without imposing disproportionate tax burdens.

For the rest of this paper, Section 2 develops the tax system first conceptually and then mathematically. Next, several model economies are constructed to quantify the characteristics of the tax system. In Section 3, the parameters of the tax system are explored, leading to a set of parameters that maximize after-tax income for the vast majority of the population with minimal government dependency. In this scenario, it turns out that the *ETR* takes the shape of a “V”, where the middle class enjoys near zero effective tax rates. This V-signature inspires the name victory tax, which has been adopted because of its general applicability to widely varying economies. The victory tax system is then compared with a flat and linear progressive tax system. In Section 4, the benefits of a simple tax structure are discussed, as well as possible public policy decisions and future work. The conclusions of this paper are given in Section 5. The main conclusion is that the victory tax as a holistic income tax system is constrained in such a way that, with minimal government involvement, households at all levels of income can reap benefits by using this tax system selfishly to gain wealth, which broadly helps society achieve a higher standard of living.

## 2. Model and Methods

The proposed tax system is deduced from five guiding principles that form the basis of a mathematical framework. This section starts with the conceptual framework, in which the rationale for the guiding principles is discussed. Recognizing that there are different perspectives, each principle is rationalized by questioning whether it has universal appeal. The goal is to transcend political biases as much as possible by rejecting what is not universal. An effort is made to distill an income tax system into essential elements. A mathematical framework is then developed with parameters that encapsulate a family of income tax systems that differ by the parameter values set by the state of the economy. This allows governments to adapt to society’s needs over time.

### 2.1. Conceptual Framework

In a bottom-up approach, five guiding principles for a holistic tax system are first listed. The rationale for each principle is then discussed as subsections. These principles shape the tax system by imposing constraints, which leads to a tax system that is easily parameterized within a mathematical framework. The principles are listed as:Income of all types is taxed at the same rate, independent of the income level.A single tax rate adjusts to ensure government fiscal stability.Government transfer is only used to establish a minimum standard of living.Three types of tax deductions incentivize wealth accumulation.(a)A basic deduction to offset living expenses.(b)Itemized deductions that promote better standard of living.(c)Capital loss deductions that promote economic growth.Tax deductions cannot be applied on government transfer.

#### 2.1.1. Income of All Types Is Taxed at the Same Rate, Independent of the Income Level

Taxing one type of income differently from another creates social discrimination. For example, it could be argued that income from work done by a teacher should be taxed twice as high as income from work done by a welder. Intrinsic to this argument is that the value of the work of a welder is more important than a teacher, or perhaps simply to offset the risks inherent in welding. However, an objective truth of this differentiation is not self-evident, as many teachers would argue otherwise. Many of these comparisons can be made, all of which end with arbitrary conclusions. Logic suggests that it is not the place of government to judge the intrinsic value of income beyond the definition of legal and illegal activities. In practice, assigning different tax rates to different types of income creates a complicated system that leads to endless debate, because there is no universal truth for all cultures, all types of economy, all types of government, and certainly not a constant in time. The same logic is true when it comes to distinguishing income from labor versus investments or other forms of income, such as gifts, winnings, insurance or inheritance.

For free-market economies, the amount of income a person generates in terms of salary or return on investments determines the value society attaches to occupation and investment. Income is determined by tangible factors, such as supply and demand, investment decisions, the wealth potential of occupations, and the desire of an individual to be wealthy. For example, the income of a surgeon can be higher or lower than a professional athlete, depending on various factors. Therefore, the allocation of different tax rates on different types or income levels must be rejected. A household with orders of magnitude more income than another will pay proportionally that much more tax, which does not discriminate on income levels. Although sale taxes can coexist with this principle, no other form of taxation of a person’s income is allowed. For example, this means that in the US, the separate payroll tax must be eliminated, as there can only be one tax rate on income, which is comprehensively taken care of by the income tax system within a holistic approach.

#### 2.1.2. A Single Tax Rate Adjusts to Ensure Government Fiscal Stability

A pertinent question is: What should the single tax rate be? A constant value (say 9%) could be argued as optimal, but this value is not self-evident. Indeed, any specific value would not be universally optimal for all cultures, economies, governments, and for all time, as the state of the economy fluctuates over time. Therefore, a variable tax rate that adapts to the revenue needed to cover projected government spending is required. A dynamic tax rate allows a government to control fiscal stability while adapting to short- and long-term economic conditions. Cycles of high and low tax rates induce an elastic response to balancing budgets (with limited liability), which ensures reliability in public services and mitigates the accumulation of long-term debt. Both of these attributes are necessary for long-term stability of society.

It is worth pointing out that policy makers have responsibility for developing debt accumulation or reduction plans. Regardless of the directions that policy makers decide, the adjustable tax rate makes tax revenue collecting responsive to government policies. For example, if more funding is appropriated toward infrastructure or defense, the single tax rate will increase, and society can monitor and judge the benefits for increased taxes. In summary, a single tax rate offers transparency in government spending, and in combination with other social-economic measures, the value that the government attaches to society’s well-being becomes transparent.

#### 2.1.3. Government Transfer Is Only Used to Establish a Minimum Standard of Living

For what reason, if any, should government transfer be used to supplement household income? Arguably, government transfers should not be used for anything other than to help a household achieve a minimal standard of living. This principle does not exclude government support in other forms, such as tax deductions and public services. For example, government spending on entitled health care, education, or other infrastructure does not constitute government transfer. However, public services must be independent of the income level, void of any income qualification.

Providing public services to poor subpopulations is unnecessary, because the poorest households will live at the poverty line, which sets a minimum standard of living. Rather than designing social programs to help the poor, public services should be designed to help society. This paradigm shift of shared interest will ensure that social programs are of high quality. It is worth noting that public services will reduce poverty by reducing basic living costs. Conversely, the poverty line rises as free public services decrease. Importantly, this principle prohibits the use of government transfers for unemployment or retirement compensation. Consequently, policy decisions will involve common interests in diverse and large segments of the population.

It is obvious that if government transfers are used to subsidize households for anything other than supporting a minimum standard of living, an arbitrary number of good reasons to redistribute income will lead to a complex tax system that is not universal. Why, however, should government redistribute income to set a minimum standard of living? Elimination of poverty is a singular case that appeals universally because of the innate human desire to live healthy and securely with dignity. This institutes the responsibility of the government to provide the means for all individuals in society to live securely with dignity over countless generations.

In practice, the poverty line must be set to balance competing factors. The poverty line should not be set too low, because more productivity in the entire population will result if society as a whole has a functional standard of living. Conversely, if the poverty line is set too high, the tax rate will rise too high, stifling economic growth. As such, the minimum standard of living that society can tolerate sets the poverty line for households. Although the way policy makers define this poverty line is left open, it must be based on income (not savings or wealth). Government transfers supplement income to establish a minimum standard of living as a safety net. If a household starts with considerable assets and then unexpectedly finds itself without income, this household can survive at the poverty line, with basic needs fulfilled. Moreover, this household can use its savings, albeit a finite resource, to live a higher standard of living.

A consequence of having one specific reason for government transfer is that the government has minimal involvement in a free market. As another example, if a household with a large accumulation of debt suddenly loses its income, it is likely to lose its possessions if an agreement with its lenders cannot be reached. Responsibility and risk tolerances exist between lenders and households taking loans. Government transfer is used only to maintain a minimum standard of living, and this results in keeping the net amount of transfer to a minimum, and hence keeps the tax rate to a minimum.

The COVID-19 pandemic is an unfortunate example of a situation in which the victory tax system maintains a stable economy during a crisis. Households automatically receive government transfers when their income falls below the poverty line due to job loss. Because of guaranteed basic income, the debate in the US on the scope of COVID-19 relief packages would be unnecessary since government transfer creates a safety net of security. Nevertheless, financial losses from businesses and households would be expected. While many lenders would be eager to force foreclosure, other lenders would use unfortunate events as a growth opportunity to attract new (sound) customers by covering businesses from bankruptcy and households from personal losses. The free market would solve the vast majority of the problems, with government regulations perhaps requiring debt collectors to exercise patience. As jobs reemerge, low-income households quickly increase their after-tax income, avoiding long-term economic stagnation.

#### 2.1.4. Three Types of Tax Deductions Incentivize Wealth Accumulation

Tax deductions are used to reduce the tax burden on households for various reasons. For a certain amount of revenue to be collected, reducing the tax burden on a subset of households requires other households to pay disproportionately higher taxes. Of course, different tax rates applied to different income levels or types cause disproportionate tax burdens. However, even if a single tax rate is applied to all income levels and types (e.g., nominally a flat tax), tax deductions create a non-flat *ETR* dependent on household income. Importantly, different types of tax deductions produce different relative benefits for households at different income levels. For example, the basic tax deduction that offsets minimum living expenses provides proportionately more benefit to low-income households. Itemized deductions predominately help households with middle-range income. Capital loss deductions on investment losses primarily benefit high-income households. In fact, low-income households cannot capitalize on capital loss tax deductions.

Among the different types of tax deductions, a balance should be sought between benefits versus the disproportionate tax burdens created across the income spectrum of households. The structure of tax deductions should benefit society, as taxpayers seek to obtain the maximum after-tax income possible from their own interests. Tax deductions therefore offer specific redistribution mechanisms for government support to motivate households to accumulate wealth, which in turn maintains a stable and growing economy. The rationale for the basic, itemized, and capital-loss tax deductions is discussed next, while key variables for the victory tax system are introduced.

*Basic tax deduction:* A minimum income is needed to live functionally in modern society. In the past, most people could live on natural resources or farm land. Unless free public services take the place of natural resources, job loss literally becomes life threatening. Hence, a basic deduction, BD, is incorporated to cover minimum living costs for a household. Although BD is a free parameter, it is appropriately related to the poverty line. Furthermore, BD should only depend on the number of dependents in a household and the cost for necessities (which is location dependent), as its sole purpose is to offset minimal living costs in the context of a social norm.

*Itemized tax deduction:* To incentivize financial independence, optional itemized deductions are allowed. Itemized deductions offer the government flexibility in the tax code to encourage certain measures, such as buying a house, accumulating a retirement portfolio, compensating costs for professional training, education or medical needs, or making donations to charities. As such, itemized tax deductions create self-interest incentives for households to take measures that also benefit society as a whole. The net itemized deduction, ID, is incorporated into the general framework of the victory tax. The total income that can be deducted is capped at a maximum. Setting a maximum deduction prevents all households from not paying tax. Two methods are used to set the maximum total deduction. A maximum deduction, MD, and a maximum percentage, MP, of net income, NI. The total deduction, TD, allowed by a household is given by:(1)TD=min(BD+ID,MD,MP×NI)
For a household with a net income above the poverty line with no itemized deductions, its taxable income, TI is given as:(2)TI=max(0,NI−TD) ∀ NI>BD

The equation for taxable income developed thus far is easy to understand. The combined total of basic and itemized deductions cannot exceed the maximum allowed deduction, nor a maximum percentage of income. Once the total deduction, TD, is determined, it will be used to reduce net income in order to achieve the taxable income. However, if the deduction is greater than the net income, NI, then the taxable income, TI, is set to zero, as it cannot be negative. The net income will be precisely defined after capital loss deductions are considered.

*Capital loss deduction:* To encourage households to increase their wealth through investments, capital loss deductions are used to mitigate risk. It is self-evident that government cannot rescue all households from financial loss. Hence, under what circumstance, if any, should government aid households to recover lost wealth? Imagine an individual that invests $100,000 in a company, and subsequently loses this investment due to the bankruptcy of that company. Another individual buys a $100,000 painting, which is inadvertently destroyed in a fire. Should both scenarios be treated equally, or should a distinction between these two losses be made? The answer rests with public policy makers who create tax law, where the answer can range from no capital loss deduction for anything to almost everything. The concern addressed next is that if the tax burden for the wealthy is disproportionately reduced, the middle class has a much greater tax burden, as the poor contribute little to tax revenue. Therefore, to justify a capital loss deduction, it is prudent to make an analogy with government-run health care, which shares an equivalent concept of large-scale group insurance.

Capital loss tax deduction is similar to an insurance program managed by the government. Specifically, all household incomes are being taxed, but the government only aids households suffering capital losses. A greater capital loss begets more government aid. The practice of spreading investment risk across the entire population to cover only households that made poor investments is like a health insurance program. That is, sick and healthy individuals are taxed, but the government only aids those suffering sickness. The greater the sickness begets more government aid. Again, aid only goes to a subpopulation to keep people functional and productive, which is beneficial to society as a whole. Although only a subpopulation will benefit from the insurance, a priori it is unknown who will use it.

The arguments against universal health care (lack of resources, highly skewed redistribution of income and poor government management) amplify against the rationale for capital loss tax deductions. The most troubling aspect is that only households with the highest income are predisposed to benefit. Thus, it is not self-evident that capital loss deductions should be included in a tax system. Nevertheless, creating opportunities that ensure the well-being of society must be the responsibility of government. When viewed as insurance, policy makers need only debate the scope of coverage. From this point of view, there is no universal answer for the scope of capital loss tax deduction or free health care services, since a weak economy cannot support the same level of coverage as a strong economy. As such, public policy debates will ultimately affect government budgets, tax rates and poverty line. The proposed tax system is designed to support the outcome of these debates within the constraints inherent in the tax system.

Since the capital loss deduction benefits society as a growth mechanism, it is included in the victory tax system to ensure generality. Nevertheless, since only a subset of households reaps the benefits, it is prudent to limit this redistribution of wealth to prevent a higher tax burden on households with much lower income. This limitation is similar to a maximum coverage limit in an insurance policy. Note that government transfer to the poor is limited by the poverty line, and a limit to the maximum itemized deduction was also introduced. In the same spirit, a cap on capital loss deductions is set by not counting capital losses that exceed capital gains within a given year. From a consistency point of view, paying tax on income must be over the same time period regardless of the income type or income level of a household. To roll over capital losses to future years, the same time period must apply to all income types. A tax system in which taxes are due annually seems reasonable, compared to alternatives such as every four months or every four years. For the prototype tax system constructed and demonstrated in this paper, capital losses are limited to one-year windows, which correspond to taxes collected annually. Although there is a maximum deduction for capital losses per year, no lifetime limit for capital losses is set. Likewise, there are annual limits, but no lifetime limit to government transfer or itemize deductions.

The definition of net income within the victory tax system is given as:(3)NI=E+max(0,CIG−CIL)
where *E* defines earnings from employment, CIG defines capital income gained, and CIL defines capital income lost. Note that CIG includes all types of income that are not earnings from employment or government transfers. Upon inspection of (Equation 3) within a given year, the government will maximally allow a household to deduct as much capital loss as gained. The amount of risk assumed is therefore determined by the skill of the investor. If an investor incurs more capital loss in a given year than capital gains, the government will not allow this excess loss to be deducted on the grounds that the investor creates too much risk for society to absorb. This restriction strengthens a free market: investors will exercise prudent judgments to ensure that gains are greater than losses when government assistance is limited. In practice, the tax system encourages investors to focus on fundamentals and long-term investments, spreading losses over multiple years. Placing annual limits on capital loss deductions (in fact all types of tax deductions) minimizes government dependency.

#### 2.1.5. Tax Deductions Cannot Be Applied on Government Transfer

This principle is self-evident, because income from government transfers is at the expense of all taxpayers who make a productive contribution to society through earnings and/or capital gains. Note that allowing deductions on government transfer would amplify government assistance. This guideline minimizes government dependency.

#### 2.1.6. Unique Property of the Victory Tax System

To emphasize the unique properties of the victory tax system compared to other basic income guarantee tax systems, an important consequence follows when the first, third and fifth principles are combined. Since government transfer income is taxed, but deductions cannot be applied to this part of household income, a regressive *ETR* emerges as a function of income percentile for the poor. The regressive *ETR* enables low-income households receiving government aid to become self-reliant and achieve higher income levels without a welfare trap (the analog to a nucleation barrier). The formation of a low-income regressive *ETR* will become clear in the results section.

### 2.2. Mathematical Framework

The five principles examined above are now considered axioms to construct the general mathematical framework of the victory tax system. Relevant variables for a household to calculate tax liability are described in Table 1 for convenient reference.

In addition to (Equation 3) defining net income, the victory tax formulas are given as:(4)GTI=max(0,BD−NI)
(5)TD=min(BD+ID,max(BD,MD),MP×NI)
(6)TI=GTI+max(0,NI−TD)
(7)TAX=VTR×TI
(8)ATI=GTI+NI−TAX
(9)$$ETR=\frac{TAX}{GTI+NI}$$

The variables {*E*,CIG,CIL} quantify income characteristics of a household. Notice that the net income of a household can never be negative. The maximum income for a household eligible for government transfer is determined by the basic deduction, BD. The word “eligible” underscores the restriction that households with incomes above BD cannot receive government transfer. Otherwise, GTI defined in (Equation 4) covers the income deficiency of a low-income household in order to put it on the poverty line. The parameters {BD,MD,MP} define the maximum limits on allowed deductions. After the total tax deduction is determined from (Equation 5), the taxable income is calculated by (Equation 6). The victory tax rate, VTR, multiplies the taxable income of a household to obtain tax liability (Equation 7). The after-tax income is calculated from (Equation 8), which adds government transfer to net income minus paid tax. The *ETR* is defined in (Equation 9) as tax paid divided by the total income. Since no deductions can be applied to government transfer, it works out that ETR→VTR when NI→0. Although counterintuitive, the poorest households pay the highest effective tax rate among the entire population.

Within the victory tax system, VTR, depends on the target tax revenue, TTR, deduced from projected budget needs for the next year set by policy makers. In addition, BD, should be proportional to the poverty line, PL which is the income required to maintain a minimum living cost. Reflecting the state of the economy, BD will change annually to ensure that the least possible after-tax income, ATI, corresponds to PL. From Equations (Equation 3) and (Equation 4) a household with NI=0 will have a taxable income equal to BD from government transfer, and after-tax income will be given by ATI=BD−VTR×BD. This shows the insightful relationship that BD=PL/(1−VTR), which indicates that as VTR increases, the basic tax deduction increases at a higher rate. When public policy makers propose to increase VTR for building infrastructure or defense, BD will automatically increase even if the poverty line remains constant.

The two parameters {MD,MP} determine the shape of the *ETR*. In practice, MD and MP will be functions of the number of dependents in a household, denoted as *d*. For simplicity, MP is considered independent of *d*. Although MD can be set with considerable latitude through tabulation, a sound approach is to set MD proportional to PL. By assuming MD=k×PL(d), the parameterization details for MD as a function of dependents are inherited from PL(d). For the US, the poverty line for a household with a certain number of dependents is publicly available in tables [23]. More generally, an objective economic measure will be used to define PL(d). Although not considered here to keep the analyses clear, the poverty line will generally depend on location (region within a country), since all regions do not have the same cost of living.

Parameters for the victory tax system are explored in Section 3. It is found that when MP=0 and ID=0, a V-shape signature emerges for the *ETR* as a function of income percentile, with ETR=0 marking the bottom of the V. Importantly, (Equation 5) determines the allowed tax deduction after taking into account basic and itemized deductions and maximum percent of NI. Since itemized plus basic deductions increase total deduction, the term max(BD, MB) that appears in (Equation 5) is needed to enforce consistency. In particular, when MD is set below BD as an independent variable, the basic deduction is still offered, but itemized deductions are no longer allowed. However, MP can reduce the maximum allowed deduction below BD without inconsistency, because MP×NI is a competing restriction on tax deductions. Note that a flat tax corresponds to the limit MP→0, where *ETR* is constant, independent of income percentile.

As MP gradually changes from 1 to 0, the V-shape morphs into a U-shape as the “U” becomes shallower, with the minimum *ETR* increasing as VTR decreases. These shape changes create an elastic tax system [24]. At MP=0 a flat tax emerges as a special case of a victory tax system, where BD sets the threshold income in which government transfer is no longer received. A prototypical victory tax system considered in this work is defined by: MD=kPL and MP=min(r, 1), where *r* is the coefficient of variation in net income over the population. Specifically, *r* is the ratio of the standard deviation in NI to the mean NI, which serves as a convenient objective measure of the economy. The application of objective measures updates PL and *r* each year, allowing the victory tax system to respond dynamically to changes in the free market and public policy.

The basic deduction is determined in the prototypical victory tax once parameter *k* is specified together with the poverty line, PL(d). By combining equations of the victory tax system, the tax liability of a household with *d* dependents is given as $TAX=VTR\times f_{d}\left(x|ID,\;VTR\right)$ from Equation (Equation 7), and its taxable income is expressed as:(10)$$\begin{array}{cc} f_{d}\left(x|ID,VTR\right) & =\max\left[0,\frac{PL(d)}{1-VTR}-x\right]\\ & +\max\left\{0,x-\min\left[\left(\frac{PL(d)}{1-VTR}+ID\right),\max\left(kPL\left(d\right),\frac{PL(d)}{1-VTR}\right),rx\right]\right\} \end{array}$$
where *x* is the net income (replacing NI for simpler mathematical notation) and the subscript *d* denotes the fact that the poverty line depends on the number of dependents in a household. Equation (Equation 10) is the result of substituting all the relevant variables described above in the arguments of Equation (Equation 6). Note that $f_{d}\left(x|ID,VTR\right)$ depends on VTR, which is the dependent variable to be determined. In addition, VTR=TTI/TTR where TTI is the total taxable income over the population, and TTR is the total tax revenue to be collected over the population. To calculate VTR=TTI/TTR, we must have TTI, which is given by the net sum of $f_{d}\left(x|ID,VTR\right)$ over all households in the population. However, to calculate TTI, we must have VTR because $f_{d}\left(x|ID,VTR\right)$ depends on VTR. Despite this circular dependence, it is straightforward to numerically solve for VTR iteratively. Uncertainties in VTR will primarily arise from estimates in TTR that will lead to surpluses or deficits at the end of a tax year when government spending deviates from budgeted allocations. These calculations are not technically difficult. The tax agency will have all income data from previous years, and TTI, based on the latest tax records, can be calculated with all details from the tax code. However, the aim of this paper is to analyze the general characteristics of the victory tax system, rather than focus on nuanced details.

For clarity and without loss of generality, the characteristics of the victory tax system are analyzed using an average household size and with the subscript *d* suppressed. This allows TTI to be expressed through the simple function N(x), which gives the income distribution over households of the average size. Defining $N_{o}$ to be the number of such households, and p(x) the probability density function quantifying how income is distributed over these households, $N\left(x\right)=N_{o}p\left(x\right)$. Assuming all households will take the maximum itemized deduction based on (Equation (Equation 10)), a lower-bound estimate for TTI is obtained. With $f_{d}\left(x|ID_{max},VTR\right)\to f\left(x|VTR\right)$ from the simplifying assumptions, the total taxable income of the entire population is given by:(11)$$TTI=N_{o}\int_{0}^{\infty }f\left(x|VTR\right)\;p\left(x\right)\;dx.$$

### 2.3. Test Economies

The victory tax system will be characterized by a series of test economies. A test economy is modeled by the income distribution of the population and poverty line. In this paper, the characteristics of the US economy are used as a starting template and for comparison in discussing the importance of the results. However, the main interest is on investigating general trends that show how the victory tax system responds to dramatic changes in income distribution. Therefore, several test economies are considered that systematically deviate from the US economy, where the size of the middle class gradually shifts from the largest segment of society to the smallest. In particular, the standard deviation in income distribution is used to expand and shrink the middle class, which respectively decreases and increases the income gap between the low middle class and the wealthiest households. Quantitative comparisons are made between a series of test economies in which the variance in household income is systematically varied as the average household income is fixed.

#### 2.3.1. Income Distribution

Accurate modeling of income distribution over a population has received much attention [3,4,5]. The income distribution of a population is represented as a probability density function (PDF) denoted as p(x), where *x* is net income. Income distributions are modeled by the κ-generalized statistics [3,4,5] and log-normal statistics. Although the log-normal PDF is a qualitatively adequate model for free markets, it underestimates population density with very high or very low incomes. The κ-generalized PDF provides a more accurate model description. In particular, the empirically observed Pareto power law tail [25] is recovered for high incomes (ultra-rich) and also more statistical weight is given to the extreme poor (both effects take away statistical weight from the middle class). The κ-generalized PDF is defined as:(12)$$p\left(x\right)=\frac{\alpha \beta {\left(\frac{x}{\mu }\right)}^{\alpha -1}\exp_{\kappa }\left[-\beta {\left(\frac{x}{\mu }\right)}^{\alpha }\right]}{\sqrt{1+\kappa^{2}\beta^{2}{\left(\frac{x}{\mu }\right)}^{2\alpha }}}$$
(13)$$\beta =\frac{1}{2\kappa }{\left[\frac{\Gamma \left(\frac{1}{\alpha }\right)\Gamma \left(\frac{1}{2\kappa }-\frac{1}{2\alpha }\right)}{\kappa +\alpha \Gamma \left(\frac{1}{2\kappa }+\frac{1}{2\alpha }\right)}\right]}^{\alpha }$$
(14)$$\Gamma \left(y\right)=\int_{0}^{\infty }t^{y-1}\;e^{-t}\;dt$$
(15)$$\exp_{\kappa }\left(y\right)={\left(\sqrt{1+\kappa^{2}y^{2}}+\kappa y\right)}^{\frac{1}{\kappa }}$$
The two parameters {α,κ} are adjusted to fit to empirical data. Although it is not prohibitively difficult to evaluate the κ-generalized PDF and other properties from κ-generalized statistics [5], the form of this distribution is not convenient to create a systematic series of test economies. Log-normal statistics are therefore used to quantitatively analyze systematic trends in a number of economies that range from a strong to a weak middle class.

The simpler log-normal PDF is defined as:(16)$$p\left(x\right)=\frac{1}{x\gamma \sqrt{2\pi }}\exp\left[\frac{-{\left(\ln x-\lambda \right)}^{2}}{2\gamma^{2}}\right]$$
(17)$$\gamma =\sqrt{\ln\left[1+{\left(\frac{\sigma }{\mu }\right)}^{2}\right]}$$
(18)$$\lambda =\ln\left(\mu \right)-\frac{\sigma^{2}}{2}$$
Again, only two parameters {λ,γ} characterize the PDF. However, they easily relate to the mean, μ, and standard deviation, σ, of income of the population. The mean household income is defined by the total net income from all households, divided by the total number of households. Both log-normal and κ-generalized distributions use the same mean household income, but they require different standard deviations to best fit the empirical adjusted gross income data for the US, as explained below.

#### 2.3.2. Test Economy Parameterization

The IRS data [26] from 1979 to 2009 was invoked to mimic the income distribution of the US. The IRS method of statistical weighting normalized the data to produce an effective number of dependents per household, and the reported income was based on 2009 dollars after adjusting for inflation. Year 2003 is chosen as an illustration. The US economy had just recovered after a market correction and began to grow over the next four years. The snapshot of the 2003 economy (in 2009 dollars) reflects a time of stability and the beginning of economic growth. The reported IRS data deals with adjusted gross income (AGI), which is analogous to net income (NI). At the level of analysis presented in this work, IRS data [26] (and NIH data [23]) are invoked to obtain realistic parameters while providing context for discussions. However, because AGI and NI are not the same, the test economies constructed in this work are best viewed as hypothetical examples.

The best fit for the IRS 2003 AGI data using the κ-generalized distribution yields α=1.50 and κ=0.56 with a relative fit error of ±5% across all income brackets. This result gives a market income distribution (synonymous with AGI) compared to total income, which includes government transfer and other forms of US subsidies, such as food stamps, etc. Henceforth, this market income distribution will be associated with the economy A, shown in Figure 1. At the far right end of the tail, 10 households with an average income of approximately $113,000,000 per year are captured. The mean household income over the entire population is $75,300 with a standard deviation of $171,712. The median income is $44,612 indicating that ultra high-income households skew the distribution, causing the mean to be considerably larger than the median. As Figure 1 shows, the most probable market income (the mode) is approximately $10,000 per year.

The log-normal distribution, which fits well to the 2003 IRS income data, has the same mean income and a smaller standard deviation of $114,475. This log-normal distribution is employed as another test economy, henceforth referred to as economy B. A smaller standard deviation in market income indicates a larger middle class, since more households picked at random are likely to be closer to the mean income. Differences in market income between economy A and B are shown in Figure 2 on a log–log plot. The κ-generalized and log-normal distributions describe similar market incomes between $4000 and $2,000,000, but the peak in the log-normal distribution shifts upward near $12,000 to compensate for depleting probability from extreme income ranges. The Lorenz curves for these two test economies, shown in Appendix A, appear virtually identical.

A series of six log-normal economies, all with the same mean income of μ=$75,000 is also considered, with standard deviations spaced by approximately powers of 2, such that σ= $7000, $14,000, $28,600, $57,240, $114,475 and $229,000 where the second largest standard deviation is economy B. In Appendix A, the six distributions are compared on a log–log plot, and their Lorenz curves are compared in Appendix A. All six log-normal economies have equal total adjustable income generated by the population, enabling a systematic method to study the dependence of a tax system on the strength of the middle class.

#### 2.3.3. Poverty Line, Tax Revenue and Government Transfer

Other information characterizes an economy, beyond income distribution, such as the number of people in the population and the poverty line. For the US economy in 2003, the number of tax-paying households was approximately $112,100,000 and the population among those households was approximately 290 million, yielding an effective household size of approximately 2.6 people. For illustration, and without altering the conclusions of the analysis, the poverty line for a mean household size of 2.6 is estimated by interpolating data from the 2009 Department of Health and Human Services data [23], yielding $16,814.

Unlike the victory tax system, which is the only source of government transfer to a household, there are many state and federal aid programs for the poor in the US to offset housing, energy, food, education, health care, and so on. Once other forms of government assistance are eliminated, a better estimate for the poverty line is $22,306. Examples are juxtaposed with both estimates to quantify how much tax burden increases when the poverty line is raised, which takes into account removal of public services for low-income households. In the victory tax system, there can be public services, but without income qualifications. As another point of reference, the poverty line in 2021 is listed as $21,960 for a household of 3 [23].

The target net tax revenue is set at $854,182,445,000 for all test economies, which corresponds to tax revenue collected by the IRS in 2003 after all government transfers were distributed. To compare all tax systems and economies, the total government transfer is calculated and added to the target net tax revenue of this hypothetical budget. When the income distribution has (more, less) households below the poverty line, the total tax revenue, *TTR*, to be collected will (increase, decrease). It is worth noting that approximately 35 percent of US federal tax revenue was derived from payroll tax, and 19 percent came from other sources [27]. Payroll tax and all other forms of individual taxation at the federal level outside of individual income tax are eliminated in the victory tax system. Although corporate tax can coexist with a victory tax system, no corporate tax is considered for simplicity of analysis in this paper. Here, all tax revenues come from individual income tax, making the *VTR* of the test economies an upper bound.

For comparison, it is insightful to examine how US tax revenues were redistributed in 2003. Total tax revenues collected were $1,952,929,045,000, of which $1,098,746,600,000 was then redistributed to households through government transfers. As such, more than 56% of the tax revenue collected was redistributed to tax payers, as captured in Appendix A. Despite the enormous amount of tax revenue redistributed to households, unfortunately approximately 15% of households live in poverty in the US [23]. In the US, tax revenues are redistributed to households at all income levels, including high-income brackets, creating both complexity and inefficiency.

## 3. Results

The adaptive nature of the victory tax requires VTR to be calculated each year. The procedure to calculate VTR from (Equation 10) favors tax revenue surpluses by assuming that all households take the maximum itemized deduction allowed. This situation yields the maximum VTR. Note that VTR must increase as itemized deductions increase to generate the same target revenue. The source of surplus is households that do not utilize all their itemized deductions. For analysis purposes, comparisons are made for a household that takes all or no itemized deductions to establish bounds for relevant quantities, such as *ETR* and *ATI*. Quantities are usually expressed as a function of the percentile of households. Percentile of households is calculated by ranking all households by net income, and then counting numbers of households at or below a certain income level to obtain a normalized scale from 0% to 100%.

### 3.1. Parameter Exploration

The shape of *ETR* as a function of the percentile of households depends on parameters {BD,MD,MP}, which are first explored as independent variables to elucidate how they affect the tax structure. Note that MP defines a percentage of total income, while BD and MD are in dollar amounts. However, when convenient, BD and MD will be specified in terms of the percentile of households. When stated BD= 20% and MD= 50%, this means BD and MD are respectively set to the income level at the 20 and 50 percentile of households. For example, for economy A, these percentiles translate to BD= $16,331 and MD= $44,612, which are the dollar amounts used in Figure 3.

Illustrated in Figure 3, as MP increases from 0 to 100% more deductions are allowed, leading to a gradual change in shape from a flat horizontal line to a “U”-shape. In Figure 3A, a flat tax appears when MP=0, causing ETR=VTR. In general, *VTR* is the y-intercept of the *ETR* plots, and ETR<VTR for households with NI>0. For MP values of 0%, 30%, 60%, 90%, the respective *VTR* are 11.97%, 15.49%, 18.35% and 20.37%. The maximum deduction implies ETR→VTR from below for high-income households. Note that *ETR* is regressive for low-income households until government transfer is discontinued. This point occurs at an income level equal to the basic deduction, which creates a kink in the *ETR*. Thereafter, a flat *ETR* applies to all households until the maximum income a household can deduct is equal to the basic deduction. At this point, a bifurcation can occur where households can use itemized deductions. The red and blue lines correspond to taking only the basic deduction versus the maximum deduction. The last segment of percentile of households has a progressive *ETR*. As MP increases, more deductions are possible, causing *VTR* to increase and the range for a flat *ETR* to decrease. Only at MP=1 will ETR=0 at the bottom of the dip. The same analysis for economy B results in the same qualitative behavior as shown in Appendix A.

As shown in Figure 4, when MP=0 the flat segment of *ETR* goes to 0%, starting at BD and ending at MD for households that take the maximum deduction. The red line tracks the maximum *ETR* for households without itemized deductions. As the basic deduction rises from 10% to 40% in steps of 10%, the *VTR* is 18.72%, 20.90%, 24.47% and 29.44%, respectively. The same analysis for economy B results in the same qualitative behavior as shown in Appendix A.

In Figure 5, *VTR* and average *ETR* are plotted for five different BD values, as a function of MP and MD. Panel A shows that *VTR* increases as more tax deductions are allowed. Panel C, on the other hand, shows that as more deductions become available, the average *ETR* decreases, where the average *ETR* reaches a minimum when MP= 100%. The flat plateaus observed in panels B and D, which extend longer for greater BD, appear because the maximum deduction cannot be less than the basic deduction. Generally, a low average *ETR* is the result of the greatest tax relief for the middle class. This result suggests that MP should be set to 100 percent for economies with large net income variations. The same qualitative behavior is shown in Appendix A for economy B.

In Figure 6, panels A and B, respectively, plot the minimum and maximum *ATI* as a function of percentile of households for different values of BD. Figure 6C compares the minimum and maximum *ATI* to the *ATI* from a flat tax. Since more tax revenue is needed for government transfer when BD increases, VTR increases too. The advantage of increasing BD is that lower-income households receive considerably more *ATI* as BD increases. However, the gains in *ATI* decrease as household income increases until a point where *ATI* decreases for high-income households. As shown in Figure 6D, households that take the maximum deduction beyond the 80 percentile have less *ATI* compared to a flat tax. Clearly, tax deductions are paid for by the progressive nature of the victory tax on high-income households (especially ultra-high-income households). Appendix A shows the same qualitative behavior in economy B.

Since BD=PL/(1−VTR), and *PL* can be quantitatively measured, MD is the only parameter left to determine. A balanced approach must compromise the desire for a generous maximum itemized deduction compared to the desire for a low *VTR*. As living costs decrease, PL will decrease, suggesting that the maximum deduction should not be large, as purchasing power is strong. Conversely, the maximum deduction should increase with an increase in living costs to ensure that itemized deductions have a positive impact on households. This leads to a prototypical victory tax system in which the maximum deduction is proportional to the basic deduction, where MD=kBD. The choice of an effective value of *k* is examined in Figure 7. The minimum *ETR* is shown in Figure 7A,C when the poverty line is at $16,814 and $22,306 while considering three values for *k*. Summarizing many of these calculations, Figure 7B,D show the average *ETR* and *VTR* as a function of *k*. Taken together, k=2 is a good compromise in tax structure. Moreover, for k<2.5 the results are insensitive to PL. The same qualitative behavior is observed in Appendix A for economy B.

The question of how the prototypical victory tax system responds to extreme variation in the economy is addressed by considering a series of six log-normal test economies, characterized by a Gini index ranging from 0.05 to 0.72 (see Appendix A). Figure 8A,C show the minimum *ETR* for these test economies, with poverty lines of $16,418 and $22,306, respectively. As the middle class expands, the *ETR* flattens, resulting in the lowest possible *VTR*. This flattening occurs because MP=min(r,1). For the 6 test economies from highest to lowest Gini index, the coefficient of variations (i.e., r=σ/μ) are, respectively, 304%, 152%, 76%, 38%, 19% and 9%. As income dispersion increases, *VTR* increases, making *ETR* very low for middle-class households. Next, Figure 8B plots *VTR* and the average *ETR* for the test economies as a function of Gini index. In Figure 8D the ratio defined by the total government transfer for eradicating poverty to the total tax revenue collected is plotted against the Gini index. For a Gini index of 0.56 (modeling the 2003 US economy), the total government transfer amounts to 23.50% or 42.30% of the total tax revenue collected when the poverty line is $16,418 or $22,306, respectively (recall 56% was used in 2003 from IRS data).

Based on the above exploration of parameters {BD,MD,MP}, henceforth the prototypical victory tax system will have: BD=PL/(1−VTR), MD=2PL and MP=min(σ/μ,1). The objective measures of the economy dynamically alter the victory tax structure, where it becomes flatter as the middle class becomes stronger. When the middle class shrinks as dispersion of net income increases, the victory tax increases *VTR* and lowers *ETR* for the middle class as it morphs into the V-signature. The steepness of V increases as the dispersion in net income increases. A steep regressive tax at low incomes provides the poor with the means to move upwards into the middle class. The progressive segment of the victory tax on high-income households provides the additional tax revenue necessary to form the V. These results show that the victory tax system is a type of governor to maintain a strong middle class.

### 3.2. Flat, Linear Progressive and Victory Tax Comparisons

Here, the victory tax is compared with a flat and linear progressive tax under identical conditions (e.g., the same income distribution, total tax revenue and poverty line). The flat tax is a victory tax with MP=0. A linear progressive tax system is defined when *ETR* is a linear function of the percentile of households with ETR=0 for a household with no self-generated income. The progressive tax rate, PTR, sets a maximum tax rate adjusted to collect the desired amount of tax revenue. For example, households at 20 and 50 percentiles will have ETR=0.2PTR and ETR=0.5PTR, respectively. Note that the linear progressive tax system does not satisfy the guiding principles 1 and 4. The *ETR* for each tax system is shown in Appendix A for poverty lines $16,814 and $22,306, respectively. For a poverty line of $22,306, the flat tax rate, FTR, is 14.80%, while PTR is 18.60% and *VTR* is 27.74%, whereas the population average *ETR* is 14.80%, 9.30% and 11.76%, respectively.

Comparisons of *ATI* in Figure 9A show that each tax system ensures the lowest possible *ATI* is at *PL*, but this occurs at different percentiles and with different trends. The linear progressive tax creates a welfare trap. That is, starting with no employment and full dependency on government transfer, the initial *ATI* is above *PL*, and then *ATI* decreases as employment increases until ATI=PL, and thereafter *ATI* starts increasing. Clearly, 30% of the lowest income households are better off not working than to work for any amount of time in a low-wage job. Although the flat tax does not penalize part-time employment and/or working low-wage jobs, it does not offer any advantage for individuals to work in a low-wage job. Quite clearly, the victory tax provides an incentive for low-income households to seek employment, where they will gain significant wealth as they move into the middle class. Consider, for example, two households living on the poverty line, the first fully dependent on government transfers and the second fully self-supporting. The second household enjoys $8565 more *ATI* (a 38% increase) due to the basic tax deduction.

For each tax system, a Lorenz curve is shown in Figure 9B for tax revenues. The Gini index for these Lorenz curves quantifies how uniform the tax burden is across income levels. A proportionate tax burden will show a similar Gini index for tax revenue as that for income received. For economy A, the Gini index for income received is 56% (see Appendix A). A flat tax without taxing government transfers yields an identical Gini index of 56% for tax revenues. However, with government transfers taxed, the flat tax gives a Gini index of 49% on tax revenues, with a higher tax burden on low-income households. The linear progressive tax system has a Gini index of 68% for tax revenues, which puts the greatest tax burden on high-income households. The victory tax reduces the tax burden on high-income households with a Gini index of 62% on tax revenues. Thus, a victory tax maintains a reasonably balanced tax burden over all income levels, where the flat tax is a special limit of the victory tax. The Lorenz curve for the victory tax is generally not monotonic, showing that low and high-income households inherit the greatest tax burden, which is why a low *ETR* is possible for the middle class.

For a (flat, progressive, victory) tax system, revenue from (7.7%, 3.3%, 5.2%) of households with the highest income corresponds to (35.2%, 30.7%, 42.3%) of the total tax revenue redistributed as government transfer. Clearly the victory tax reinvests the most back into society, which is responsible for removing the welfare trap [18] through its regressive *ETR* for low-income households. In general, as government transfers decrease due to an expanded middle class, the degree of disproportionate tax burden decreases. Appendix A shows the same qualitative behaviors with a lower poverty line.

Now the question of how these three tax systems respond to the poverty line is addressed. Tax rates as a function of *PL* for the flat, linear progressive and victory tax systems are shown in Figure 10A–C, respectively. This analysis is applied to the series of six log-normal economies described in Section 2.3.2 to elucidate the effect of income dispersion for a fixed mean income. Here, the characteristic tax rate of a tax system is tracked as the poverty line is varied for different income dispersion scenarios. The lowest dispersion of $7000 represents the situation in which the vast majority of households fall into the middle class, with only rare cases of poor or rich households. As income dispersion increases, the middle class shrinks, and the percentage of extreme poor significantly increases. In contrast, only a small increase in ultra-rich households occurs. At high income dispersion, wealthy households pay much more tax revenue, because most of the taxable income generated across the population comes from wealthy households. Moreover, as income dispersion increases, more households fall below the poverty line, which increases tax revenues that must be collected to cover the increase in government transfer. As a reference point, the final value of the considered standard deviation ($229,000) is approximately twice the dispersion found in the 2003 US economy. Although income dispersion in the US economy has steadily increased since 2003, it is still lower than the largest dispersion considered here.

In the victory tax system (including the special case of flat tax), it is seen that *VTR* increases as *PL* increases. In particular, as the poverty line is increased, *VTR* must increase more rapidly, as population dispersion is larger. Consequently, a large middle class keeps the *VTR* low, and the progressive part of the victory tax on high-income households will be most shallow.

Figure 10B shows that a linear progressive tax system requires an increase in PTR as the middle class expands. Interestingly, PTR is generally insensitive to the poverty line. Only when there is a wide income gap will the dependence on the poverty line become substantial. As the middle class shrinks, the tax rate rises quickly, but unfortunately there is no mechanism for the middle class to reexpand. Contrary to the linear progressive tax, as shown in Figure 10A,C, the flat and victory tax have the intuitive rank ordering that *VTR* decreases as the strength of the middle class increases.

For economy A shown in Figure 10D, *VTR* rapidly increases as a function of *PL*, while the average *ETR* remains markedly insensitive to *PL*. A low average *ETR* can be obtained even when *VTR* is high, because most of the population substantially benefits from itemized tax deductions, except for households for which the maximum itemized deduction is dwarfed relative to their net income. As the middle class shrinks, the tax base also shrinks, and the *VTR* increases. This creates an elastic response to income gaps, which means the tax structure makes it easier for the middle class to expand when market pressures act to shrink it. Conversely, when the middle class expands, the V flattens, encouraging accumulation of wealth in households above the middle class level. In summary, the victory tax has desirable elasticity [24] to create stability in the economy, where the sharpness of the V-shape increases as income dispersion becomes extreme, and flattens as the middle class expands, fueling a consumer-based economy.

### 3.3. Distribution of Government Assistance

Government transfers and tax deductions are two forms of government assistance. However, the deductions on capital losses were not reflected in the above analyses, as net income distributions account for capital gains and losses. To quantify the degree of government assistance in terms of optional deductions, estimates of the average amount of itemized and capital loss deductions must first be modeled. Simple models are presented to quantify the total itemized and capital deductions in the population as a function of the percentile of households. These models are not part of the victory tax system. However, they are useful in characterizing how government assistance is distributed across households. The qualitative estimates made in this subsection are only intended for illustration and discussion purposes.

The net itemized deduction, *NID*, is modeled as: $NID=\frac{f^{2}}{2}\max\left(0,NI-BD\right)$ where *f* is the percentile of households. For economy A with a poverty line of $22,306, the basic deduction is $30,871 and the maximum deduction is $44,612. The net itemized deduction for a household at the (20, 40, 60, 80) percentile, with corresponding NI of ($16,331, $33,878, $57,787, $101,844), produce a mean itemized deduction of ($0, $240, $4844, $22,711). The factor of $f^{2}$ models the qualitative trend that the more income a household has, the more likely it will utilize itemized deductions up to MD.

The total capital loss, *TCL*, is modeled as: $TCL=NIf^{4}CL/\left(1-CL\right)$. The factor of $NIf^{4}$ models household capital gains, where it rapidly tends to zero as f→0, and tends to NI as f→1 to reflect the empirical observation that high-income households have larger portions of their income from investments. The parameter CL is a ratio of capital losses to capital gains for the tax year. Since CL reflects an average over the population, the range from 0.1 to 0.7 suffices to quantify the fraction of government assistance applied to capital loss deductions. Illustrating this qualitative model: For CL= 20% and for households at the (20, 40, 60, 80) percentiles, *TCL* is modeled as ($7, $216, $1877, $10,429). For example, a household with gross income of $112,273 and capital loss of $10,429 has a net income of $101,844.

Employing the models for NID and *TCL*, government assistance, GA, is given as:(19)GA=GTR×(1−VTR)+VTR×(TD+TCL)
where *TD* is total allowed deduction from basic and itemized deductions. In addition to government transfer, reducing the tax burden on households through tax deductions contributes to government assistance. The average tax deductions made by households at the (20, 40, 60, 80) percentiles are estimated to be ($16,338, $31,329, $37,592, $55,042). The average of (TD+TCL) over the population gives the average tax deduction, which can exceed MD because *TCL* is included. Due to *TCL*, a plot of government assistance versus percentile is not informative, because government assistance to the top 0.1% of households dwarfs all other forms of assistance on an absolute scale. For example, with CL= 20%, the average tax deduction at the 99.9 percentile is $436,583. However, relative comparisons in government assistance can be made. Here, GA is divided by GTR+NI to define a fraction of assistance that shows how government assistance is distributed over the percentile of households. Note that the assistance fraction for the extreme poor is not 100% for the victory tax. For a household with NI=0, which receives all its income from government transfers, the assistance fraction is equal to GA/GTR=1−VTR. For economy A, and *PL* of $22,306, recall *VTR* is 27.74%, meaning 0.7226 is the fraction of government assistance when NI=0. The relative government assistance decreases if only basic and itemized deductions are considered.

The percentage of government aid as a function of percentile of households for different CL ranging from 10% to 70% is shown in Figure 11A,B. The deviation of CL considered causes “fanning” in the tail of the assistance fraction. Qualitative characteristics and general trends are not sensitive to the models used for tax deductions. High-income households with high profit-to-loss ratios have the smallest assistance fractions, but they receive much more absolute assistance than poor households. Middle-class households have too much income to benefit from government transfers, and too little income to get much government assistance from tax deductions. However, in the victory tax system, middle-class households have the lowest *ETR*. As the poverty line increases, the assistance fraction increases for households at all income levels, not just low-income households. Together, these trends allow the victory tax system to be relatively balanced, without creating serious disproportionate tax burdens. In short, the extreme poor and ultra-wealthy pay the highest tax rates, but also receive the greatest government assistance through various mechanisms.

Figure 11C,D show Lorenz curves for government assistance. The associated Gini index of these Lorenz curves will be 0 for proportionate government assistance at all income levels. It is noticed that the Gini index depends on the average ratio of population loss to profit. For (10%, 30%, 50%, 70%), the Gini index for government assistance is (0.00, 0.17, 0.34, 0.53). The relatively low Gini index values indicate an adequate balance in the way government assistance is distributed across income levels. Since the Lorenz curve is mostly below the diagonal line (when CL> 10%), ultra-high-income households receive the most government assistance on an absolute scale. This results suggest the ultra-rich will significantly increase government dependency in economic downturns, where massive losses occur (i.e., CL large). This is another sign of elasticity to maintain social-economic stability, because government aid to the wealthy is strongest at the time when the risk of investing is highest.

## 4. Discussion

In the US, there is a bad connotation when discussing the poor taking advantage of the welfare system, or the rich taking advantage of tax breaks. With the five guiding principles satisfied, all taxpayers are encouraged to take advantage of the victory tax system for personal benefit, regardless of their station in life. As taxpayers take advantage of the victory tax system based on rational self-interest, a synergic benefit for society as a whole arises from the constraints on the tax structure. The mathematical framework enables a pragmatic approach to taxation, because competing interests within the population are represented within a small parameter space, which helps funnel opposing public policy options into transparent objectives. For illustration, several issues that are important to the US are discussed herein.

### 4.1. Consolidation of Diverse Interest Groups

A victory tax system necessarily eliminates myriad assistance programs in the US that require a need-based qualification, such as food stamps, housing/energy assistance, unemployment, social security and medicare. All types of income based welfare programs are consolidated into a single mechanism for distributing government transfer based on need, without judgment qualifications. Social security is an excellent topic of discussion. Unfortunately, the US social security program often fails to meet the basic needs of the elderly, and its solvency is questionable because new revenue from the workforce is not synchronized with recipient needs. Moreover, social security benefits have expanded beyond a retirement fund tied to age.

In the victory tax system, the unemployed and retirees belong to the same interest group. A large diverse subpopulation of recipients of government transfer will lobby for public policy to overestimate the poverty line (requiring a higher tax rate), which will counteract other large diverse subpopulations that will lobby for a lower tax rate to stimulate economic growth. In addition, as the middle class lobbies for greater itemized deductions, this also requires increasing the poverty line. Notice that the vast majority of advocates for increasing the poverty line will not belong to the marginalized subpopulation living in poverty. Competing interests from large influential groups will encourage an objective measure of the poverty line to be set within public policy. It is critical for large powerful and diverse groups to argue for specific changes within a small parameter space to produce effective outcomes.

### 4.2. Legal Requirements

If proof of citizenship or legal residency is required, illegal immigrants seeking public welfare are discouraged from living in the country because they will be identified. Furthermore, it would be disadvantageous for households with incomes below the poverty line not to report income for work.

### 4.3. Tax Form Simplicity

It is envisioned that a simple form determines tax liability. The main page of the tax form will be less than 1 page long, with optional worksheets for specifying itemized deductions and capital gain/loss information. A look-up table could be invoked to determine the basic and maximum deductions depending on the number of dependents in a household. Although public policy determines the tax code, the presence of maximum deductions does not warrant complexity in the code. For illustrative purposes, an example of the first page of a victory tax form is given in Table 2.

To cover a diverse range of possibilities, Table 2 compares 12 examples of filled tax forms for a range of household percentiles from 0 to 90 in steps of 10, as well as 95 and 99. A maximum deduction of $44,612 (being twice the poverty line) together with a $30,871 basic deduction create a cap of $13,174 on itemized deductions. From line #6 on the tax form, households at and above the 80 percentile request the maximum deduction possible. Households within the percentile range from 40 to 70 pay more taxes than they need, because they are likely not to have enough surplus income. Allowed itemized deductions help society and the household. For example, an important itemized deduction should be to invest in a retirement fund. As a household achieves greater net income and/or reduce expenses, more itemized deductions for a retirement fund are possible, among other allowed reasons.

### 4.4. Financial Security from Job Loss

Regardless of a household’s savings or previous income levels, a household automatically gains a minimum guaranteed income at the poverty line if job losses lead to no income. A household with a living standard far above the poverty line would inevitably exhaust their savings due to an extended period of job loss. However, it is not the role of government to maintain differences in wealth in households, even for a short time.

### 4.5. Right to Work

The basic guarantee of income in the victory tax system eliminates the need for a minimum wage. Without a minimum wage, many companies are likely to lower wages below a living wage. Nevertheless, a low-wage job will increase *ATI* above the poverty line, as the regressive *ETR* leads to significant increases in *ATI* from modest income increases. This increase in net income was not the case with the negative income tax, which was tested in the late 1960s to early 1970s in North America [9]. Within a victory tax system, low-wage entry-level jobs can benefit society. For example, a young person with no previous work experience earning low wages increases the collective *ATI* of a household. This is a win–win–win situation: A company acquires cheap labor, a new worker gains valuable training, and the household increases its net income. The steep regressive tax encourages workers to accept low paying jobs in exchange for building skills. After gaining experience, workers should expect to move into higher-paying positions, creating rapid turnover in entry-level jobs. Companies will have to adjust their wages to balance the turnover rate with the costs of training new workers.

### 4.6. Right Not to Work

As companies become dependent on the government to pay low-wage workers, if left unchecked, this practice will inevitably develop into a modern form of slavery [28], when work is required in exchange for government transfer. Enforcing work for government transfer at the poverty level is analogous to forcing companies to pay workers a high minimum wage. To prevent exploitation of workers, work requirements cannot be applied to the basic income guarantee. In particular, the right not to work is a necessary balance in a free market in which individuals are free agents who promote their own agenda for wealth accumulation. Guaranteed basic income subsidizes both workers and owners. When workers are independent agents, the main reasons for unions become unnecessary. For example, a person can refuse to work in conditions they consider inappropriate, unworthy of their talents, too little pay, or because the job is uninteresting. This gives workers free time to develop new skills and seek better-paying jobs based on their merits. Whether a person receives government transfer because of retirement from the workforce, does not find work or decides not to work is irrelevant to the victory tax system.

### 4.7. Productivity in Society

The victory tax system promotes a productive society, not by providing comfortable financial security to people, but rather by providing incentives for people across the income spectrum to take jobs, become more demanding for higher salaries and better benefits, and to regularly make financial investments. Of course, there will be a subpopulation of people that will take guaranteed basic income and never work. A victory tax system allows the free market to determine workforce equilibrium, and it embraces the income distribution from that free market that includes non-working households. The victory tax system makes no judgments about why people work or not.

It is important to stress that productivity cannot be quantified in monetary terms, because not generating taxable income is not the same as unproductive. For example, a basic income guarantee can help low-income parents meet household needs to raise children, which is a productive activity for society. People in low-income households often have health problems that prevent them from working [15]. Some people will choose to live a life near the poverty level while providing community benefit through good deeds. Allowing people to retire at the age of their choice also eliminates the arbitrary mandatory retirement age set by government. In short, the victory tax system offers each individual the opportunity to achieve success in their own terms, which can evolve over time.

Consider the case in which a person (young or old) wishes to pursue creative interests. Such passions can be pursued productively, rather than inhibited by the need to work for mere survival without dignity. Observing peers accumulating personal wealth creates a powerful incentive for the vast majority of people not to indefinitely pursue personal interests strictly at the poverty line. Generally, wealth accumulation occurs over one’s lifetime. Consider the most likely scenario when young adults leave their parents’ home. For an inexperienced worker, it is generally difficult to find a good-paying job that meets basic needs. The victory tax system allows young adults to become independent and productive taxpayers sooner due to the regressive nature of the *ETR*. Starting with government transfer, as an individual’s economic status increases government assistance changes to itemized and capital loss tax deductions. Thus, all forms of government assistance help create a productive society as individuals capitalize on their rational self-interest.

### 4.8. Security in Personal Wealth

The victory tax system offers short- and long-term opportunities for prosperity through itemized deductions. For example, the cost of investing in stocks and bonds can be an itemized deduction. In the short term, this itemized deduction contributes to wealth accumulation by encouraging households to reduce their *ETR* and increase their *ATI* by investing in the economy. In the long run, the accumulated wealth of a household can be exploited during retirement as other income, which can substantially increase *ATI* above the poverty line for the middle class. For lower-income households, modest but significant gains in *ATI* will result from supplementing government transfer. Furthermore, the wealth generated by a taxpayer stays with the taxpayer at all times. Households can sell assets for income at any time without imposing penalties for early withdrawal or waiting until a certain retirement age.

Policy makers should allow a wide range of itemized deductions to offer opportunities for prosperity. For example, allowing itemized deductions on accumulating assets, such as a house, or to offset the cost of higher education or for training on workforce skills. Other forms of security could include itemized deductions on health insurance or health care costs. In this way, households can pay less taxes by taking measures to strengthen their financial independence and well-being. In summary, many allowed itemized tax deductions in the tax code will give households the opportunity to use surplus income for personal gains that create benefits for society.

### 4.9. Catalyst for Micro-Businesses

The victory tax system creates a supportive environment for micro-enterprises to form. As low-wage jobs are subsidized, new businesses can rely on this to reduce startup costs. For example, new businesses can form a mission to attract low-skilled workers to help the local community while building skills for their recruited workers. This paradigm replaces working in low-wage jobs without growth opportunities. With a foundation for social security, the private sector has the means to solve local community problems without high barriers. The safety net of guaranteed basic income enables low- to middle-income households to take risks in entrepreneurial endeavors that would otherwise be prohibitive. Over time, micro-enterprises can grow into larger businesses.

### 4.10. Responsible Government

The floating tax rate helps avoid runaway deficits, as it can adapt to government budgets that take into account debt and repayment plans to control the ratio of debt to gross domestic product (GDP) through public policy. It is worth noting that public policy may promote large debt accumulation to keep the victory tax rate lower, but this jeopardizes long-term stability. For the US, an open budget to the public provides transparency to determine if taxation has representation. With only a few key tax parameters, debates will focus on why tax rates, poverty lines, or deductions should be changed. Furthermore, any proposed changes to the parameters can be modeled and tested with the consequences predicted. This clarity will make public policy debates more substantive, such as evaluating the effectiveness of expanding free public services compared to increasing the poverty line.

Government transfers should be distributed continuously, as they are part of a steady income of a household. An efficient and convenient method of government transfer and collection of taxes owed is consistent with Adam Smith’s four principles [29], the third of which states: “Every tax ought to be levied at the time, or in the manner, in which it is likely to be convenient for the contributor to pay”. This logic also applies to government transfers in the victory tax system, which eliminates the need for government to administer complex assistance programs. However, there will be subpopulations in society that will refuse government support on the grounds of principle or incompetence. In the latter case, the government should support institutions that care for people who are dependent on others, such as nursing homes, institutions for mentally ill or homeless, etc. If a person must be put in a public or private institution because they cannot live independently, the institution becomes their household, and will receive government transfer in their behalf. Financial administrators of these institutions will be obliged to pay income tax on the government transfer received in aggregate form. In this way, the victory tax system supports the entire population, including the “forgotten” people in society who must live as dependents.

### 4.11. Role of Corporate and Other Taxes

In the analysis of the victory tax system, all other sources of tax revenue, such as sales and corporate tax, were dismissed. As the example economy of Table 2 shows, *VTR* approaches 30% when there is a weak middle class and highly skewed net income distribution, as is currently the case in the US. An argument often made is that low corporate taxes create GDP growth. For the analyses given here, corporate taxes are 0% across the board, regardless of the size of the company/business. Remember that in 2003, approximately 19 percent of tax revenues came from other admissible sources. Therefore, it is feasible to shift *VTR* down by approximately 5% when other tax revenues are taken into account. Ignoring all other admissible tax revenue sources, the 27.7% *VTR* is not prohibitively high, demonstrating that the victory tax system is cost effective. Discussion of an appropriate corporate tax system goes beyond the scope of this paper.

### 4.12. Future Work

This work can be expanded in several ways. The consequences of a victory tax system should be quantified by large-scale agent-based modeling to simulate an evolving economy [30]. Different initial economic conditions under different public policy constraints should be systematically investigated and compared with other tax systems. A few key questions should be addressed: Does the victory tax system stabilize the middle class? What is the impact on market income distribution? How will GDP be affected? Other features to be explored include how to deal with spatial (regional) inhomogeneity, how to synergistically combine this system with corporate taxation, and how to transition from an existing tax system into a victory tax system. The victory tax system, along with other tax systems, should be tested in economies that will be largely automated. For those interested in further testing or developing the victory tax system, a C++ program that is used to generate all the results presented here (including other types of Lorentz curves that are not shown) is available in Appendix A. Although many aspects and consequences for universal basic income have recently been addressed [8], these findings, albeit insightful, are not directly applicable to the victory tax system.

## 5. Conclusions

Five guiding principles have been developed for a pragmatic income tax system that generates revenue for the government while providing basic welfare, and encourages households across the income spectrum to accumulate wealth. These principles limit the way in which government redistributes wealth, but the details of implementation are left to public policy. These principles lead to a mathematical framework describing a family of tax systems based on a small parameter space where the parameters are informed by the state of the economy. A prototypical victory tax system was created that shows the feasibility of a holistic paradigm for a poverty-free and productive society without disproportional or excessive tax burdens. To highlight the effect of the constraints imposed by the mathematical framework, four key features of the victory tax are summarized.

Every year, a victory tax rate (*VTR*) is set to cover projected government spending based on public policy, which plays a role in incurring or reducing debt. The *VTR* increases when government spending increases, and vice versa.As the maximum allowed deduction increases, the *VTR* increases, but the effective tax rate (*ETR*) for the middle class remains markedly low, and possibly zero, due to its V-shape as a function of household income percentile. Conversely, as the maximum allowed deduction decreases, the V-shape of the *ETR* becomes shallower and gradually flattens until a flat tax appears when no deductions are allowed, resulting in the lowest possible *VTR*.The combination of government transfer and a basic deduction creates a regressive tax for low-income households and guarantees a basic income for all households, which is set at the poverty line after taxes are paid. By taxing government transfers at the *VTR*, no welfare trap is formed, creating a substantial incentive for households to generate income. The poverty line sets the minimum standard of living that society can tolerate, which depends on availability of public services. An increase in the poverty line will increase the *VTR*, and vice versa. Each year, the poverty line and income dispersion are objectively measured and updated to keep the victory tax responsive to changes in the economy.Government assistance in the form of government transfer, basic deductions, itemized deductions and capital loss deductions create opportunity for wealth accumulation in households across the income spectrum. The *ETR* on taxable income for the ultra-rich is no higher than for the extreme poor. As the middle class shrinks, the *VTR* increases, and the *ETR* for low and high-income households become more regressive and progressive, which stabilizes the middle class.

The victory tax makes it easy for taxpayers to calculate tax liability, collect government transfer when needed, and for government to set tax rates that will generate the projected revenue to cover its expenditures. While the victory tax cannot prevent runaway deficits, the adjustable tax rate provides the government with a means to control debt, which is an important factor in setting the single tax rate. The constraints within the tax system make public policy objectives transparent, suggesting that policy debates will become meaningful for the typical taxpayer. This is because government assistance, either through direct transfers or deductions, offer incentives for households to take advantage of the victory tax system out of rational self-interest. As low-income households accumulate wealth, society’s standard of living can rise significantly, suggesting that the middle class will be the dominant segment of the population.

## Figures and Tables

**Figure 1 entropy-23-01492-f001:**
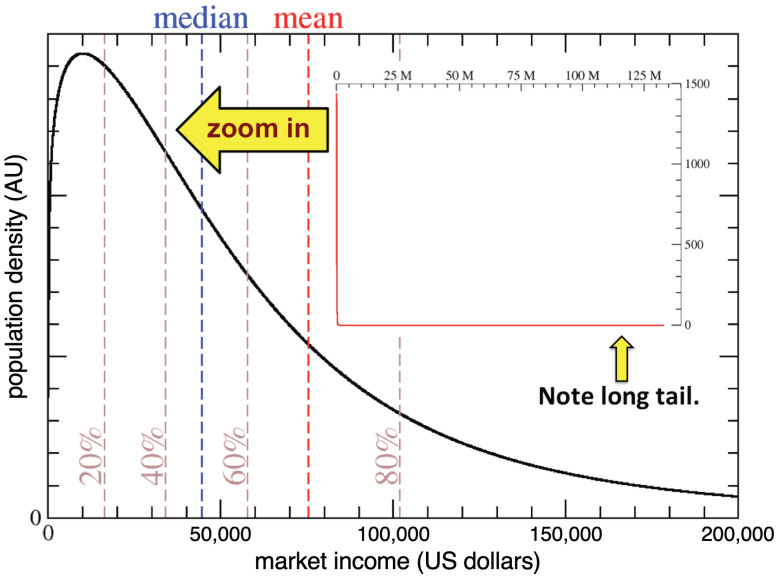
Market income distribution for economy A: The household AGI probability density ranging from $0 to $200,000 is shown. Population quintiles are marked with light brown dashed vertical lines at 20%, 40% 60%, and 80%. Median and mean incomes are respectively marked as blue and red dashed vertical lines. (**inset**) The same distribution is shown without cutting out data from high income households. The horizontal line at the bottom of the graph highlights the heavy tailed distribution, indicating only a tiny number of households reach this level of income.

**Figure 2 entropy-23-01492-f002:**
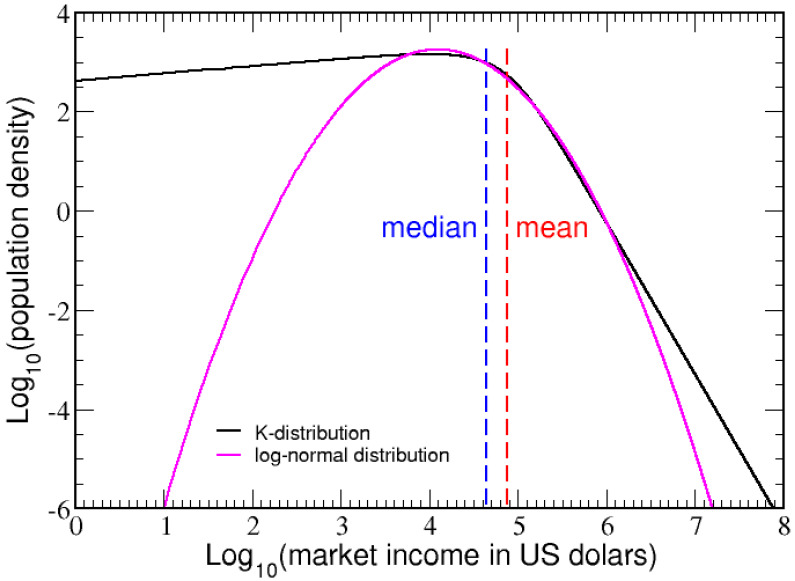
Market income distribution comparison: On a log–log scale the κ-distribution defining economy A and log-normal distribution defining economy B are compared. The κ-distribution puts more statistical weight for the ultra-rich and extreme-poor subpopulations as reflected in the wings of the distribution. The empirical median and mean incomes for the US 2003 economy are shown as blue and red vertical dashed lines.

**Figure 3 entropy-23-01492-f003:**
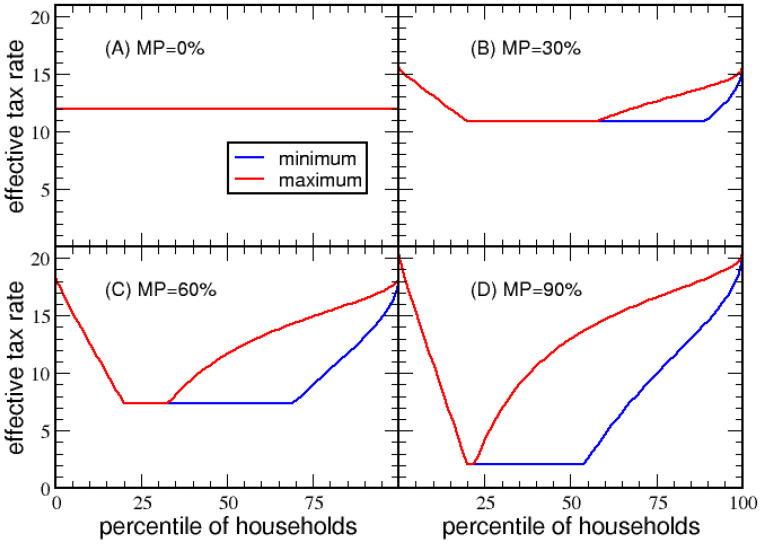
Effective tax rate comparisons: For economy A the minimum *ETR* (blue) and maximum *ETR* (red) are shown for BD= 20% and MD= 50% in each of the panels with different MP given by: (**A**) 0% (red line covering blue line); (**B**) 30%; (**C**) 60%; (**D**) 90%.

**Figure 4 entropy-23-01492-f004:**
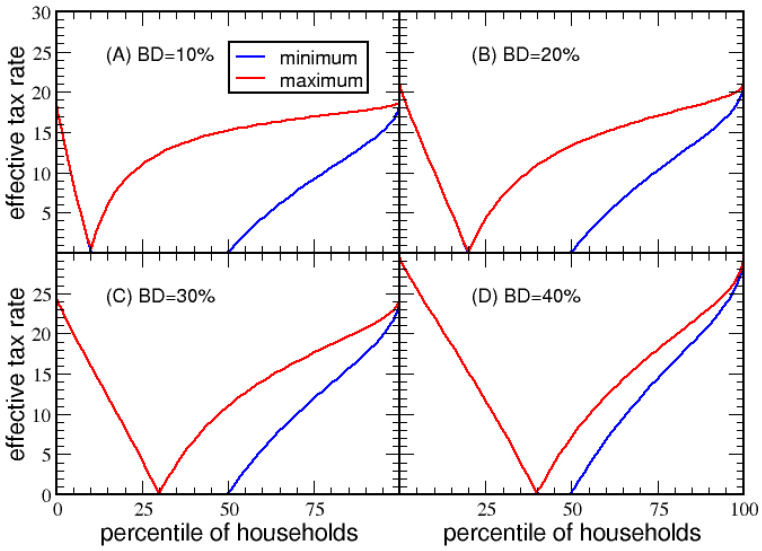
Effective tax rate comparisons: For economy A, the lower- and upper-bound *ETR*s are shown for MD= 50% and MP=100% in each of the panels with different BD given by: (**A**) 10%; (**B**) 20%; (**C**) 30%; (**D**) 40%. At large BD, the signature “V” shape appears for the maximum *ETR*.

**Figure 5 entropy-23-01492-f005:**
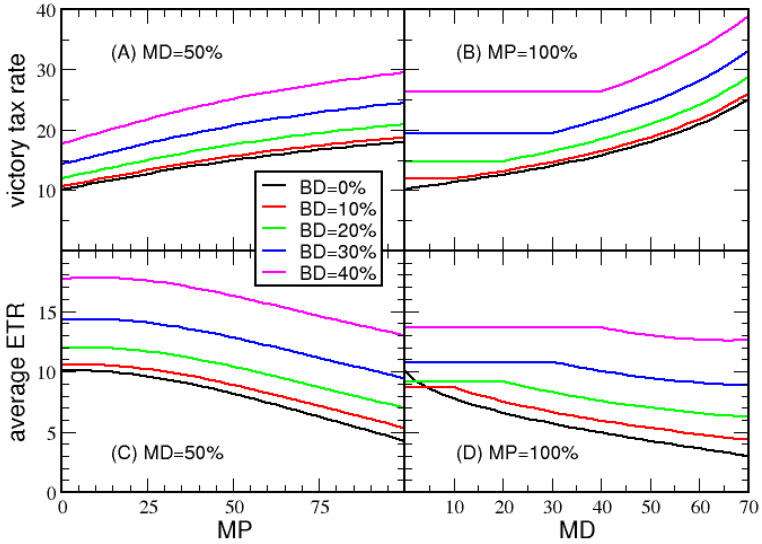
Tax rate comparisons: Trends in tax rates for economy A are explored. The legend applies to all panels, where different color lines represent a BD of 0%; 10%; 20%; 30%; 40%. The victory tax rate is shown as a function of (**A**) MP; (**B**) MD%. The average *ETR* is shown as a function of: (**C**) MP; (**D**) MD%. In panels A and C, MD=50%, and in panels B and D MP=100%.

**Figure 6 entropy-23-01492-f006:**
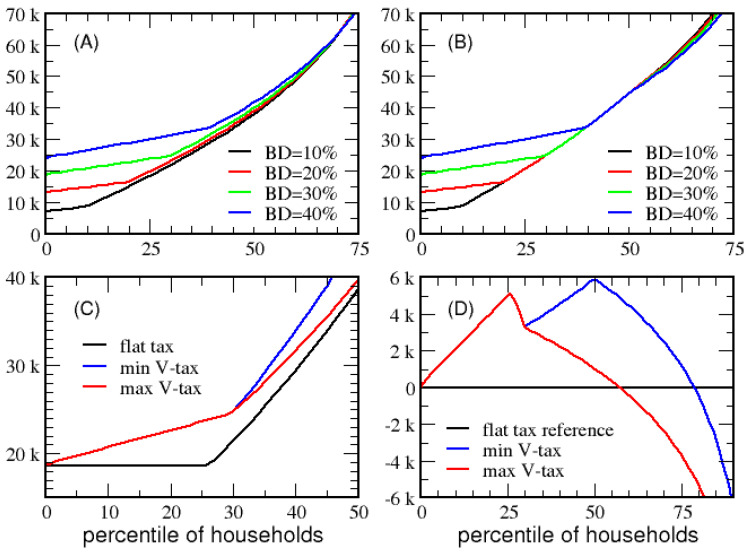
After-tax income comparisons: Trends in *ATI* are explored for economy A. With MD=50%; MP=100%, and BD ranging from 10% to 40% the *ATI* as a function of household percentile is shown for the case: (**A**) minimum *ATI*; (**B**) maximum *ATI*. (**C**) Comparing a flax tax to the victory tax (V-tax), the minimum *ATI* (max V-tax), maximum *ATI* (min V-tax) and flat tax *ATI* are shown with BD=30%, MD=50%, MP=100%. (**D**) For the same parameters used in panel C, the difference in min/max V-tax *ATI* relative to the *ATI* for a flax tax is shown.

**Figure 7 entropy-23-01492-f007:**
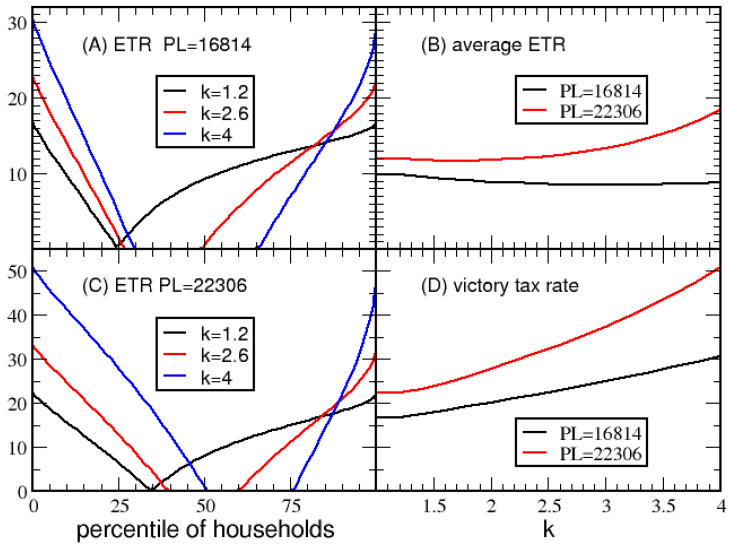
Maximum deduction exploration: For economy A, the *ETR* as a function of percentile of households is shown in panels (**A**) and (**C**) for a poverty line of $16,814 and $22,306, respectively. The maximum deduction is set to be proportional to the poverty line, where different color lines show different proportionality constants set at 1.2, 2.6 and 4. As a function of *k* and for two different poverty levels, panel (**B**) shows the average *ETR* and panel (**D**) shows the victory tax rate.

**Figure 8 entropy-23-01492-f008:**
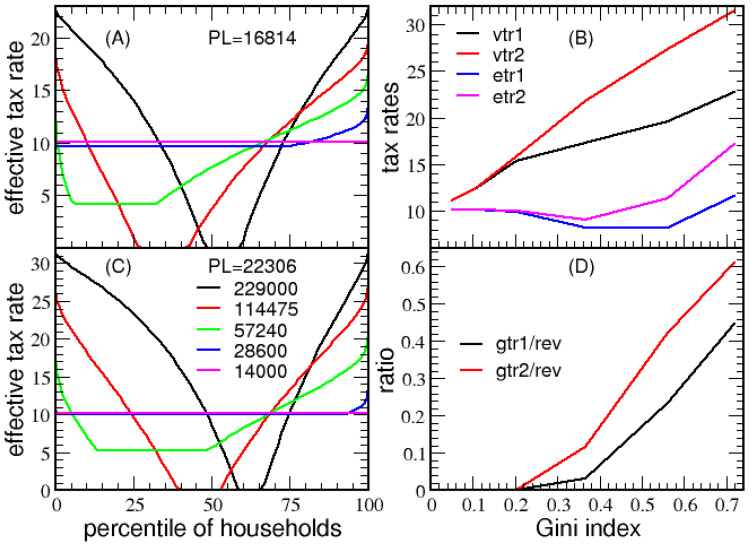
Systematic variation in net income dispersion: Trends in *ETR* with respect to income dispersion are shown in panels (**A**) and (**C**) for six economies described by a log-normal distribution with standard deviations ranging from $7000 to $229,000 about a mean income of $75,300. The panel (**C**) legend also applies to panel (**A**). The *ETR* for the $7000 standard deviation is not shown because if plotted, it is flat and hidden under the magenta line. As a function of Gini index, panel (**B**) plots the victory tax rate (vtr1 or vtr2) and average *ETR* (etr1 or etr2) for cases 1 and 2 corresponding to a poverty line of $16,814 and $22,306. For the same two cases, panel (**D**) plots total government transfer divided by total tax collected (gtr1/rev or gtr2/rev).

**Figure 9 entropy-23-01492-f009:**
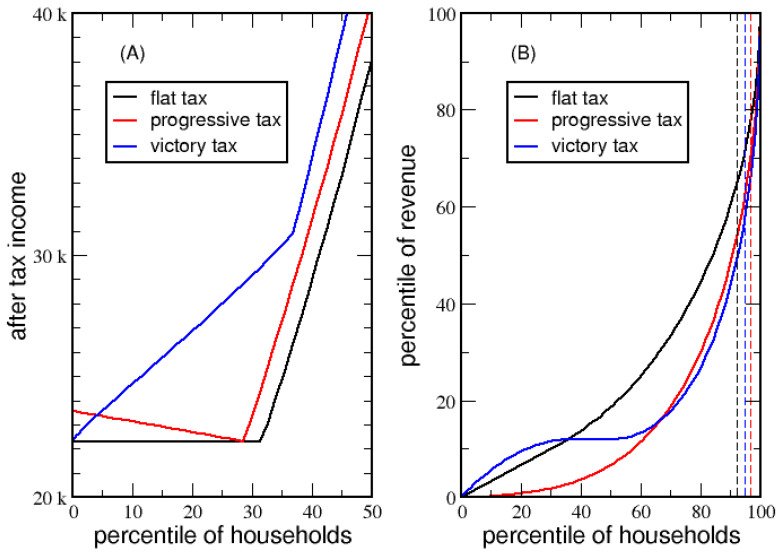
Tax system comparisons for economy A with a $22,306 poverty line: (**A**) After-tax income for flat, linear progressive and victory tax systems are shown in different colors; (**B**) The Lorenz curves for tax revenue are shown. The dashed vertical lines of corresponding color to the tax system indicate the percentile of households from which point onward the collected tax revenue is sufficient to pay for all government transfer needed to eliminate poverty.

**Figure 10 entropy-23-01492-f010:**
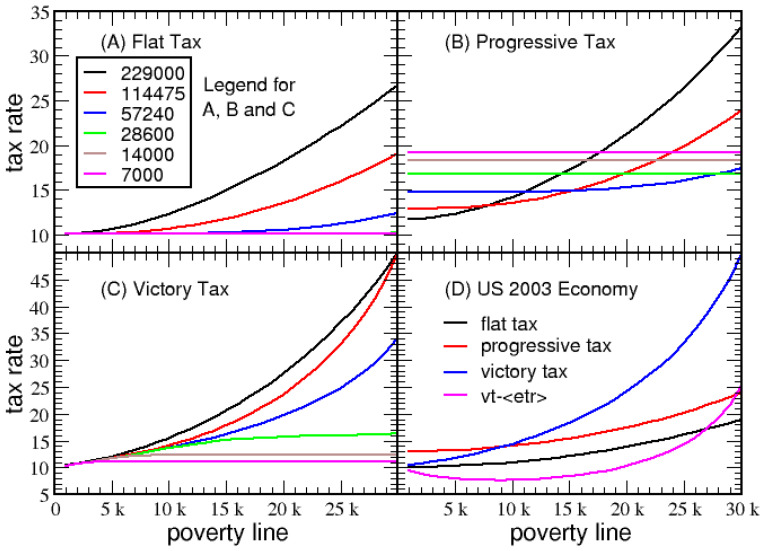
Tax rate dependence on poverty line: A series of six log-normal economies with income dispersion characterized by standard deviation ranging from $7000 to $229,000 about a mean income of $75,300. (**A**) flat; (**B**) linear progressive; (**C**) victory tax. (**D**) The tax rate for a flat, linear progressive and victory tax are shown together for economy A, which is used to mimic the 2003 US economy. The average effective tax rate over the population for the victory tax (denoted as vt-etr) is also shown as a function of poverty line.

**Figure 11 entropy-23-01492-f011:**
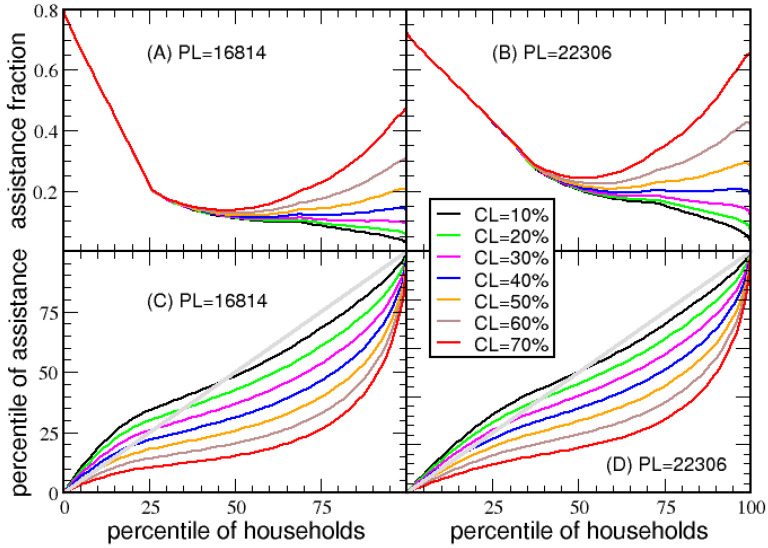
Government Assistance Distribution: Analysis of tax deductions for economy A: (**A**,**C**) consider a $16,814 poverty line and (**B**,**D**) consider a $22,306 poverty line. Panels (**A**,**B**) show the assistance fraction as a function of percentile of households. Panels (**C**,**D**) show the Lorenz curves for government assistance from (**A**,**B**), respectively. The diagonal line colored light grey in panels (**C**,**D**) is drawn to guide the eye.

**Table 1 entropy-23-01492-t001:** Alphabetically ordered list of variables for the victory tax system and their description.

Variable	Variable Description
ATI	After-tax income.
BD	Basic deduction for minimum living expenses.
CIG	Capital income gain.
CIL	Capital income loss.
*E*	Earnings through employment.
ETR	Effective tax rate.
GTI	Government transfer income to a household.
ID	Itemized deduction taken by a household.
MD	Maximum deduction allowed for a household.
MP	Maximum percent of household income that can be deducted.
NI	Net income of a household after capital loss deductions.
PL	Poverty line.
TAX	Tax liability.
TD	Tax deduction.
TI	Taxable income.
TTI	Total taxable income over the entire population.
TTR	Total tax revenue from the entire population.
VTR	Victory tax rate applied to all households and income levels.

**Table 2 entropy-23-01492-t002:** Calculation of tax liabilities for twelve exemplar households. The first column gives the line numbers on the tax form. The second column gives the instructions. The six columns afterward represent example answers for households at different percentiles, f. The first 11 rows of the table correspond to the first 11 line numbers: #1 government transfer; #2 earnings income; #3 other income; #4 deductible income; #5 basic deduction; #6 itemized deductions; #7 total deductions; #8 reduced income; #9 net income; #10 taxable income; #11 tax owed. The *ETR* is given in the bottom row. All examples are based on economy A with a poverty line of $22,306 for a household of 3 with 1 dependent. The models in Section 3.3 for itemized and capital loss deductions are used to fill table entries in lines #3 and #6.

Line #	Instructions	f = 0%	f = 10%	f = 20%	f = 30%	f = 40%	f = 50%
1	direct data entry	$30,871	$22,383	$14,540	$6228	$0	$0
2	direct data entry	$0	$8488	$16,326	$24,603	$33,705	$44,064
3	data from worksheet	$0	$0	$5	$40	$173	$558
4	add line 2 and line 3	$0	$8488	$16,331	$24,643	$33,878	$44,622
5	data from lookup table	$30,871	$30,871	$30,871	$30,871	$30,871	$30,871
6	data from worksheet	$0	$0	$0	$0	$241	$1719
7	add line 5 and line 6	$30,871	$30,871	$30,871	$30,871	$31,112	$32,590
8	subtract line 7 from line 4	−$30,871	−$22,383	−$14,540	−$6228	$2766	$12,032
9	greater of line 8 or $0	$0	$0	$0	$0	$2766	$12,032
10	add line 1 and line 9	$30,871	$22,383	$14,540	$6228	$2766	$12,032
11	multiply line 10 by 0.277443	$8565	$6210	$4034	$1728	$768	$3338
	**effective tax rate =**	27.7%	20.1%	13.1%	5.6%	2.3%	7.5%
**Line #**	**Instructions**	**f = 60%**	**f = 70%**	**f = 80%**	**f = 90%**	**f = 95%**	**f = 99%**
1	direct data entry	$0	$0	$0	$0	$0	$0
2	direct data entry	$56,289	$71,677	$93,501	$135,429	$189,913	$411,893
3	data from worksheet	$1498	$3616	$8343	$20,455	$36,958	$97,951
4	add line 2 and line 3	$57,787	$75,293	$101,844	$155,884	$226,871	$509,844
5	data from lookup table	$30,871	$30,871	$30,871	$30,871	$30,871	$30,871
6	data from worksheet	$4845	$10,883	$13,741	$13,741	$13,741	$13,741
7	add line 5 and line 6	$35,716	$41,754	$44,612	$44,612	$44,612	$44,612
8	subtract line 7 from line 4	$22,071	$33,539	$57,232	$111,272	$182,259	$465,232
9	greater of line 8 or $0	$22,071	$33,539	$57,232	$111,272	$182,259	$465,232
10	add line 1 and line 9	$22,071	$33,539	$57,232	$111,272	$182,259	$465,232
11	multiply line 10 by 0.277443	$6123	$9305	$15,879	$30,872	$50,566	$129,075
	**effective tax rate =**	10.6%	12.4%	15.6%	19.8%	22.3%	25.3%

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
