# Peer review of "Victory Tax: A Holistic Income Tax System"

_entropy, 2021, doi:10.3390/e23111492_

Round 1

Reviewer 1 Report

In my opinion, the real purpose of taxation, from the stabilization of a normal level of the middle class (and not a thinner and thinner level of the middle class all over the world), to the provision of resources for necessary government services…

The beginning of taxes in human history is given by the unique tribute or the unique tax ( e.g. the first population censuses in world history deal with identifying the number of population and the level of the tax, etc. in order to finance the army), but not with the necessary level of the army (principle of army maximization dominates and involves maximizing taxes at the limit of affordability…

I believe that a holistic vision with a sense of maximum multidisciplinary vision (in extension approach) is preceded by some realistic but also necessary methodological steps:

  1. i) an attempt to analyze intersystem charging,
  2. ii) an attempt to extend to a multisystem as wide as possible,

iii) a clarification of the limits of multisystem expansion,

  1. iv) an approach to the subsidy as an inverted or algebraic tax with a minus sign (negative), etc.

In principle, the article wants to pose a theoretical problem and deserves to be published, it is well written and valid, but I would consider it is necessary to improve it, especially the introductory part and also the publication part.

I accept after minor revision of Introduction and conclusion

Author Response

Your opinion is noted, and in a few more places, including the conclusion, I note that the poverty line depends on public services. How much public service a government institutes will depend on public policy. Public service is not incompatible with the victory tax, as mentioned early on and multiple times. As free public services increase, the poverty line will decrease.

Based on your comment, I noted the historic point that taxes were not initially considered to benefit society and provided a good reference series for the history of taxes. Thanks for this suggestion. The introduction has been edited to better explain the motivation of the work while removing some repetitive statements, and improving the grammar. The conclusion was edited extensively and expanded to summarize all the results from this work. The original version of the conclusion was too terse. 

Reviewer 2 Report

This is an interesting read that, if nothing else, reminds us that tax codes can be much simpler and effective than we typically use. While I disagree with the author's conclusion that the victory tax helps to avoid runaway deficits (it does not), I still found the article interesting in its commitment to demonstrating a tax code with very few governing parameters. The article is necessarily vague in its treatment of itemized deductions (this is an obvious quagmire for actual policymaking), although it does clearly make that case that allowing for larger deductions generates a higher overall tax rate.

As I noted above, what it doesn't do is constrain deficits. By implicitly assuming that tax rates are set based on "target tax revenue", the model allows the government to essentially borrow endlessly to fund the difference between actual government spending (not captured here) and actual revenue net of total transfers. Thus, it suffers from the same problem as the existing tax structure in the U.S. although it would be much simpler and have much lower administrative costs. That's reason enough to read the paper. In fairness, I'm not sure how you could build in an effective balanced-budget requirement (other than literally requiring it), and this would effectively handicap the government from using fiscal policy as a macroeconomic stabilizer, in the conventional sense.

Author Response

The reviewer has provided the answer to why the victory tax system does not require government to enforce a balanced budget or place a maximum cap on how much debt it can accumulate. In fact, initially I wanted to build this in as a guideline, but could not. It must be a matter of public policy, and exceptions need to be allowed even if the de facto policy was to maintain a balanced budget. I read through the manuscript very carefully, and modified wherever necessary so that now, in my opinion, it does not imply the victory tax system will prevent runaway debt. I emphasize this goal would be a matter of public policy, and that the victory tax system can be an effective tool to prevent runaway debt, if that was the policy. Furthermore, I directly acknowledge what you say in the conclusions as well.

Reviewer 3 Report

In the manuscript, the authors propose the victory tax system which emerges with three parameters that determine a minimum allowed tax deduction, a maximum allowed itemized deduction, and a maximum deduction defined by income percentage. The simplicity of the victory tax makes it easy for taxpayers to calculate tax liability and helps public debates focus on pragmatic solutions that make public policy objectives transparent. As taxpayers at every income level benefit from the victory tax through financial stability and wealth accumulation, society’s standard of living increases. I thus believe the paper meets the criteria for publication on Entropy. However, in order to accept this manuscript for publication on Entropy. I would like to ask the authors to address the following minor concerns:

[Comment 1]

There are some grammar mistakes. Careful proofreading should be performed.

[Comment 2]

In the section Mathematical Framework, the authors list the relevant variables for a household to calculate tax liability. In order to understand the meaning of these variables more clearly for the readers, a table is suggested to organize them.

[Comment 3]

In Equation (10) on page 10, some symbols are not explained in detail. It is suggested that more detail about your mentioned formula in this paper should be given.

[Comment 4]

In Figure 10, the authors give the analysis of the dependence of the tax rate on the poverty line, and a series of six lognormal economies with income dispersion characterized by standard deviation ranging from $7000 to $229000 about a mean income of $75300. The authors should give detailed reasons why the standard deviation is set at this interval.

[Comment 5]

The authors should carefully check the missing important references, such as"Entropy 2017".

Author Response

Comment 1: A slow and careful read over the document by myself and a colleague was performed. Several edits where made throughout the document.

Comment 2: Thanks for the suggestion, table 1 was created to define the variables in a more organized way and space saving way compared to the original layout.

Comment 3: Thanks for the suggestion. I expanded the discussion around and about equation 10, which helped clarify the calculations considerably.

Comment 4: Thank you for your attention to detail. I added a considerably long discussion about the standard deviation range and its significance to this section. This helps elucidate the results.

Comment 5: I must confess that I was unaware of Entropy-2017 special issue. However, I did check it out and looked through all the contributions. Only the paper by Tsallis had some relevance, but since that was a review paper, and only a small part of it mentioned wealth distribution (and not taxes), I felt there was no good connection to this work for referencing any of those papers. I already referenced an entire book dealing with the subject matter regarding income and wealth distributions.